# The Regional Hydro-Ecological Simulation System for 30 Years: A Systematic Review

**Benxin Chen [1,2], Zhifeng Liu [1,2,*], Chunyang He [1,2], Hui Peng [3], Pei Xia [1,2] and Yu Nie [1,2]**

[1] Center for Human-Environment System Sustainability (CHESS), State Key Laboratory of Earth Surface Processes and Resource Ecology (ESPRE), Beijing Normal University, Beijing 100875, China; benxin@mail.bnu.edu.cn (B.C.); hcy@bnu.edu.cn (C.H.); xia_pei@outlook.com (P.X.); 201921051129@mail.bnu.edu.cn (Y.N.)

[2] Faculty of Geographical Science, School of Natural Resources, Beijing Normal University, Beijing 100875, China

[3] Key Laboratory of Marine Environment and Ecology, Ministry of Education, Ocean University of China, Qingdao 266100, China; pengh@ouc.edu.cn

\* Correspondence: zhifeng.liu@bnu.edu.cn

**Abstract:** As the Regional Hydro-Ecological Simulation System (RHESSys) is a tool to simulate the interactions between ecological and hydrological processes, many RHESSys-based studies have been implemented for sustainable watershed management. However, it is crucial to review a RHESSys updating history, pros, and cons for further improving the RHESSys and promoting ecohydrological studies. This paper reviewed the progress of ecohydrological studies employing RHESSys by a bibliometric analysis that quantitatively analyzed the characteristics of relevant studies. In addition, we addressed the main application progress, parameter calibration and validation methods, and uncertainty analysis. We found that since its release in 1993, RHESSys has been widely applied for basins (<100 km$^2$) within mainly seven biomes. The RHESSys model has been applied for evaluating the ecohydrological responses to climate change, land management, urbanization, and disturbances, as well as water quality and biogeochemical cycle. While most studies have paid their attention on climate change, the focus has shifted to the application for land management in recent years. This study also identified many challenges in RHESSys such as the inaccessible data and parameters, oversimplified calibration approach, few applications for large-scale watersheds, and limited application fields. Therefore, this study proposed a set of suggestions to overcome the limitations and challenges: (1) Developing a new approach for parameter acquisition and calibration from multi-source data, (2) improving the applicability for a large-scale basin, and (3) extending the scope of application fields. We believe RHESSys can improve the understandings of human–environment relationships and the promotion of sustainable watersheds development.

**Keywords:** RHESSys; ecohydrology; watershed; landscape sustainability

## 1. Introduction

Regional Hydro-Ecological Simulation System (RHESSys), a distributed physical processes-based ecohydrological model, has been developed mainly by a two-way coupling of ecological models and hydrological models. RHESSys has a hierarchical structure with three main modules that allow hydrological, microclimate, and ecological processes to be simulated separately in different layers and to reflect the multi-scale feature of watersheds [1,2]. By parametrizing regional eco-hydrological processes through a series of coupled physical mechanism models at different levels, RHESSys can model the interactions between hydrological, climatic and ecosystem processes in a watershed,

and thus, can simulate the regional multi-components cycle of nitrogen, carbon, and water [2]. In particular, RHESSys has been applied for various research fields due to the characteristics of the hierarchical structure and coupled multiple physical processes. First, the hierarchical structure defined by multiple processes improves the simulation efficiency as the multiple processes are operated individually at multiple spatio-temporal scales. In addition, RHESSys incorporates a plant physiological process that simulates the carbon and nitrogen cycling of vegetation and soil to reflect the nonlinear ecosystem response [2]. Moreover, RHESSys has a flexible structure to be coupled with other models (e.g., WMFire, phenology models) [3–5] and consequently is adapted for various research fields. In summary, RHESSys has been designed to assess the interactions between vegetation and water for ecohydrological research and sustainable management of watersheds by simulating regional carbon, nitrogen, and water cycles and distributions.

Over the last 30 years, RHESSys has been continuously advanced in model structures and algorithms, and applied for various basins to support local water resource management [6–8]. For example, Zabalza–Martinez et al. [8] applied RHESSys to simulate hydrologic responses to climate and land-use change scenarios for a basin controlled by the Boadella–Darnius Dam in Spain and suggested water resource management strategies for the reservoir and corresponding sub-basins. Peng et al. [9] evaluated the impacts of soil and water conservation measures on the runoff of the Jinghe basin in China for not only filling gaps of local assessment of the effectiveness of soil and water conservation measures but also supporting watershed management. Martin et al. [6] also simulated the water yield of the Yadkin-Pee basin in North Carolina, USA to evaluate the impacts of climate change and human activities for reasonable water resource management. As many studies have shown the advantages and limitations of RHESSys, a systematic evaluation of the application progress of RHESSys can provide useful and scientific information for the in-depth understanding of the strengths and weaknesses of the RHESSys model. Furthermore, such a systematic review may help in improving models and optimizing the application of RHESSys [2].

This study aims at reviewing the progress of RHESSys-based research. Firstly, we introduced the basic structure, principles, and development history for the RHESSys model. By a systematic review of relevant literature, the progress of RHESSys-based research was summarized including the calibration approaches, verification methods, uncertainty analysis, and applications. The ultimate objective of this paper is to reconsider the structure, principles, main research topics, and future trend of RHESSys for further support of the improvement and even broader application of RHESSys.

## 2. The Basic Structure and Development History of RHESSys

Since Band et al. released the initial version of RHESSys in 1993 [1,10], RHESSys has become a matured and popular ecohydrological model (Figure 1) over the last 30 years. Initially, RHESSys was designed by explicitly coupling the Forest Biogeochemical Cycles (FOREST-BGC) canopy model [11] with a Mountain Climate Simulator (MT-CLIM) [10], and advanced by coupling with a topography based hydrological model (TOPMODEL) [12] for the hydrologic process. This version of RHESSys is capable of simulating water, carbon, and the nitrogen cycle in a forest-dominated basin. The Forest-BGC model can simulate vegetation growth, nutrient, and water cycle of the forest ecosystem while MT-CLIM mainly conducts interpolating meteorological variables at a climate station to target points. TOPMODEL is a physical-based quasi-distributed hydrological model. In the coupled RHESSys, the simple soil–water module in the FOREST-BGC was replaced by the vertical infiltration and soil flow process in TOPMODEL [2]. In the updated version of RHESSys, the Forest-BGC was replaced by Biome-BGC to simulate the eco-hydrological processes of multiple ecosystems, while it was coupled with a soil–plant nutrient cycling model (CENTURY$_{NGAS}$) [13] to optimize the simulation process of the nitrogen cycle, especially nitrification and denitrification. For an advance in hydrological processes, RHESSys incorporated an explicit hydrologic routing model (DHSVM) that could account for non-grid-based patches and nonexponential transmissivity profiles. More details on each module and algorithms of RHESSys have been described by model developers [1,2].

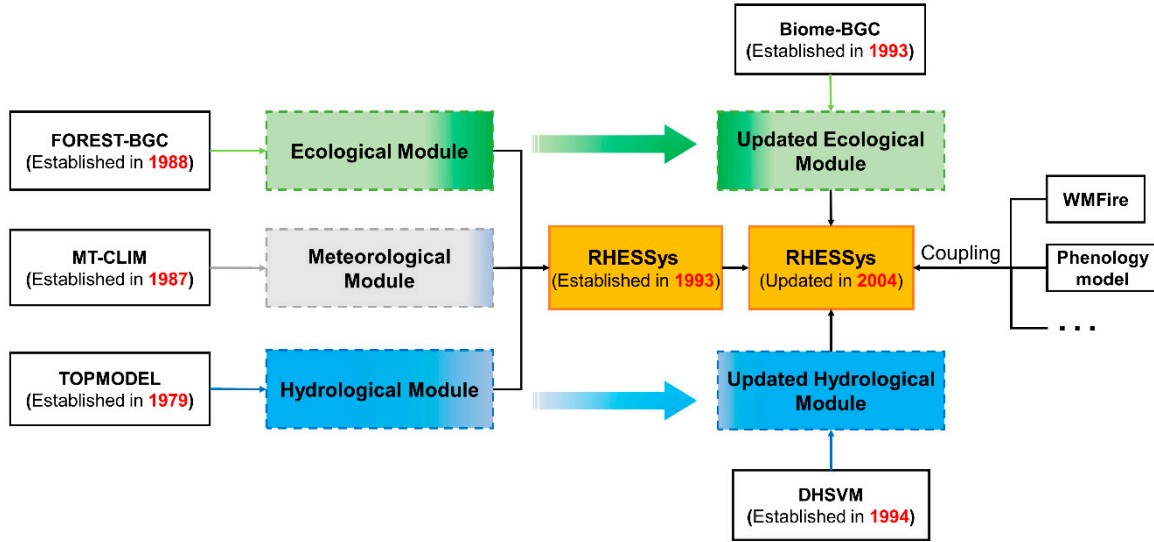

**Figure 1.** The development history of RHESSys.

In addition, RHESSys provides a useful tool to simulate the surface processes of watersheds by coupling with other models (Figure 1). For instance, the impacts of fire on the ecohydrological process were evaluated by loosely coupling a fire spread model (WMfire) with RHESSys [3,4]. Moreover, some studies examined the effects of phenology changes on the watershed runoff and evapotranspiration process by coupling a phenology model [5].

RHESSys describes a basin as an object containment hierarchy of Basin, Zone, Hillslope, Patch, and Canopy strata, which allows different hydrological and ecological processes to be modeled at different scales [14] (Figure 2). The patch represents the smallest unit that has similar soil moisture and land cover. The soil representation is a relatively simple bilayer generalization, i.e., unsaturated and saturated layers. Vertical soil moisture processing and soil biogeochemical cycles are modeled at this level considering snowpack and litter stores. Patches can be derived by multiple layers of land use, soil moisture distribution, or topographic map. Canopy strata describes the vertical process above the ground at the same resolution and partition with the patch, which mainly refers to the physiological processes for plants such as respiration and photosynthesis. In short, the subsurface process is modeled in the patch while the aboveground process is modeled in the canopy strata. The hillslope defines horizontal water movement and redistribution between patches to produce streamflow in a sub-catchment that drains into a stream reach. The hillslope is often derived by GIS-based terrain-partitioning algorithms. The zone defines a region that is usually partitioned by the distribution of climate stations or elevation bands. The zone contains meteorological variables and uses extrapolation methods to characterize spatial variation in these variables. The basin defines a spatial boundary for a catchment, which generally refers to the entire watershed simulated by the model. The basin typically aggregates the net flux of water, carbon, and nitrogen across the whole study area [2,14].

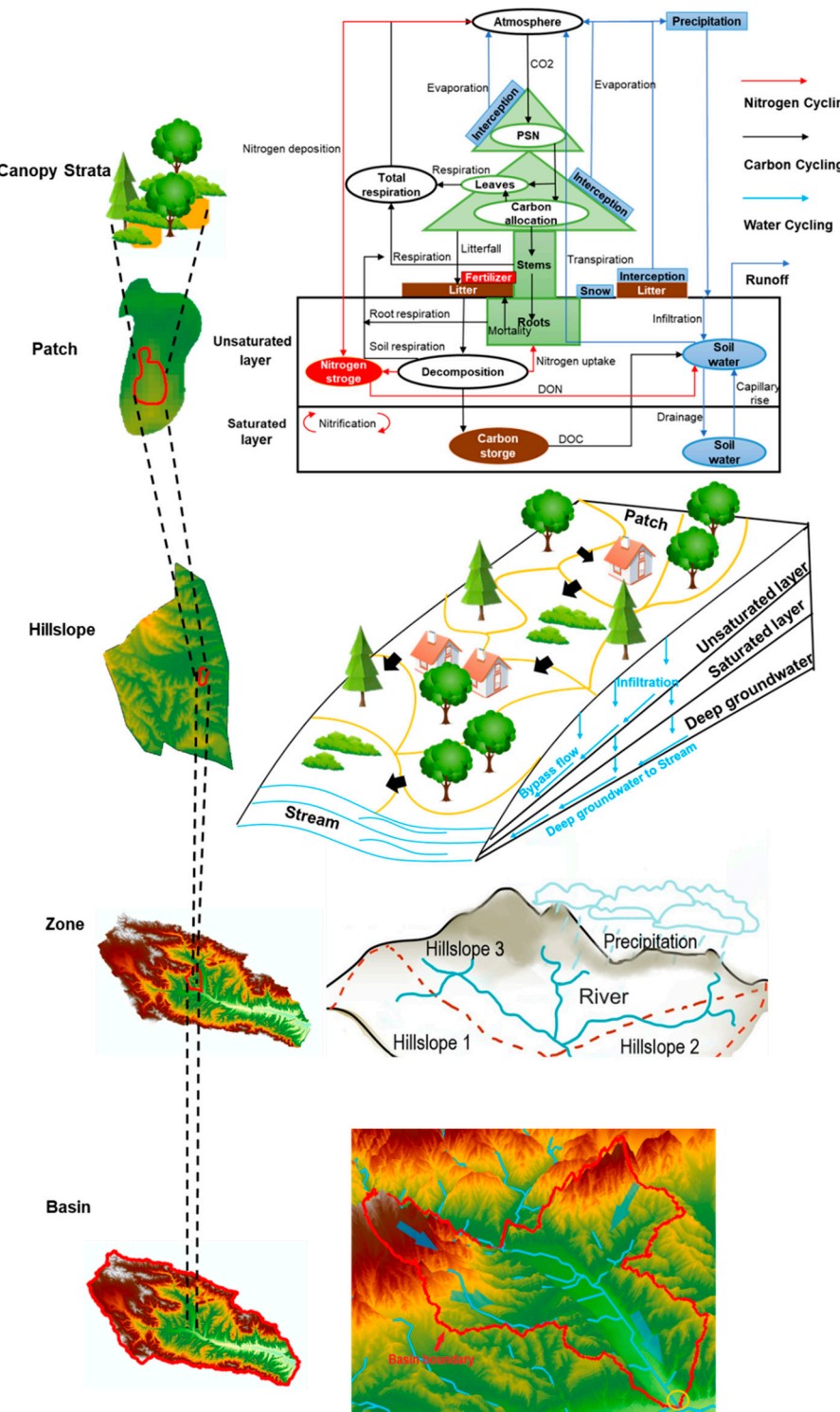

**Figure 2.** The hierarchical structure of RHESSys (Edited based on [15,16]).

Table 1 illustrates structures, key processes, and applications for various ecohydrological models. It is prominent that RHESSys has a hierarchical structure (Figure 2) to better reflect the multi-scale characteristics of ecohydrological processes in a watershed while other ecohydrological models have 'basin-grid' and 'basin-subbasin-grid' structures. For instance, Gorelick et al. [17] found that RHESSys can handle mixed and heterogeneous land cover at a fine spatial resolution and is suitable for more detailed ecohydrological modeling in small catchments. However, SWAT (Soil and Water

Assessment Tool) is spatially lumped at the subbasin level and applies to model large basins with spatially well-segregated landscapes. Moreover, RHESSys realized the bidirectional coupling of the eco-hydrological processes, depicting not only the effects of soil water processes on the plant physiological processes, but also the impacts of vegetation growth on hydrological processes. In contrast, most earlier ecohydrological models (i.e., SWAT and TOPOG_IRM) simplified the complicated vegetation–water interactions. Morán–Tejeda et al. [18] compared the performance of RHESSys and SWAT with the same input and application areas. The results suggested that RHESSys was more sensitive to land cover and vegetation change while SWAT produced larger changes under climate change. The major underlying cause was that SWAT uses empirical functions of potential evapotranspiration to calculate evapotranspiration, but RHESSys estimates evapotranspiration in a more process-based way, as a complex representation of canopy transpiration controlled by rooting depth, stomatal conductance, etc. Therefore, RHESSys has the advantage for watershed simulations that focus on land cover or vegetation–water interactions. In addition, RHESSy has a flexible structure to further dynamically couple with other models such as phenology, fire, and land-use models, leading to a wider range of applications to support the water resources management under assorted conditions [2–5].

**Table 1.** Characteristics of RHESSys and ecohydrological models used in literature.

| Model | Structure | Key Processes Representing Eco-Hydrological Interactions | Applications | References |
|---|---|---|---|---|
| RHESSys | Basin-Zone-Hillslope-Patch-Canopy strata | Carbon and nitrogen cycling of soil and vegetation, Plant physiological process, Evapotranspiration, Lateral flow, Slope confluence | Urbanization, Water quality, Climate change, Disturbance, Water resource management, Land management, Biogeochemical cycle | [2] |
| TOPOG_IRM | Basin-subbasin | Carbon cycling of vegetation, Plant physiological process, Evapotranspiration, Lateral flow, Slope confluence | Climate change, Disturbance, Water resource management, Land management, Biogeochemical cycle | [19] |
| SWAT (Soil and Water Assessment Tool) | Basin-subbasin-Hydrological response units | Evapotranspiration, Lateral flow | Urbanization, Water quality, Climate change, Water resource management, Land management | [20] |
| BEPS-TerrainLab (Boreal Ecosystem Productivity Simulator-TerrainLab) | Basin-grid | Carbon and nitrogen cycling of soil and vegetation, Plant physiological process, Evapotranspiration, Lateral flow, Slope confluence | Water resource management, Biogeochemical cycle | [21] |
| tRIBS-VEGGIE (TIN-based Real-time Integrated Basin Simulator-Vegetation Generator for Interactive Evolution) | Basin-tin | Carbon cycling of soil and vegetation, Plant physiological process, Evapotranspiration, Lateral flow, Slope confluence | Disturbance, Water resource management, Biogeochemical cycle | [22] |

## 3. Research Trends and Characteristics

Our systematic review followed the steps proposed by Khan et al. [23]. We used "RHESSys" as a keyword for literature retrieval in the Science Citation Index Expanded (SCI-Expanded) and Social Science Citation Index (SSCI) databases of the Web of Science Core Collections. In addition, a full-text search was carried out on Google Scholar with "RHESSys" as the keyword. The search period was from 1990 to 2019, and the retrieval date was 13 January 2020. According to the above search criteria, 1059 records were found. These records were further filtered according to the following

criteria: (1) published in SCI or SSCI journals, (2) applied RHESSys in a certain area. The full text of these papers was read, and 90 literature related to RHESSys were identified for a systematic review (Table S1). Even our literature search was not inclusive considering some relevant studies potentially published in other journals; our major findings are robust in terms of the general trends revealed. Based on the selected literature, the research trends, the main application progress, methods for calibration and validation, and the future perspectives of RHESSys were summarized.

Figure 3 shows the number of published literature and their citations related to RHESSys since 1990. Since the papers related to RHESSys were first published in 1993, the number of published papers and citations has been steadily growing. By the end of 2019, a total of 90 papers had been published and cited 3324 times.

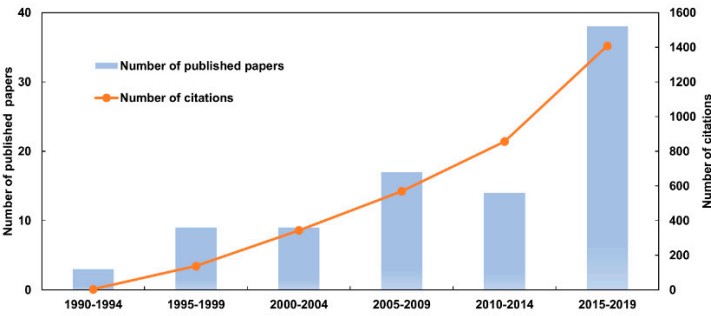

**Figure 3.** Number of published papers and citations.

To visualize the main content of the RHESSys applications, we conducted a word cloud analysis based on the title, abstract, and keywords of the collected 90 papers (Figure 4). After filtering out irrelevant keywords, "Water" occurred the most frequently. Other keywords such as "Climate" "Forest" "Soil" "Hydrology" "Watershed" "Ecosystem" and "Streamflow" appeared at a high frequency, representing the primary simulated objects. The frequency of "Change" "Simulate" "Process," and "Increase" was also high, indicating the dynamic simulation process of RHESSys.

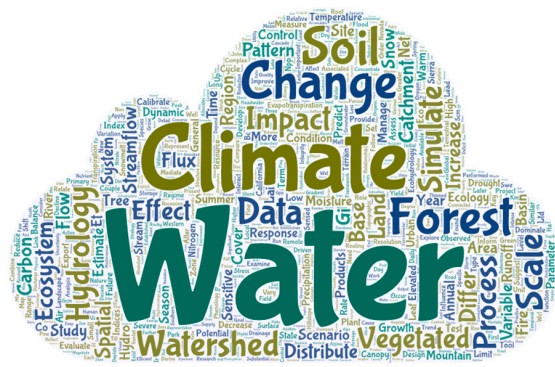

**Figure 4.** Topic words of RHESSys-related papers. Note: Extracted topic words are based on the collected 90 papers (title, abstract, and keyword). The font size is proportional to the relative frequency of each word; a word with larger font represents more frequently occurring in papers; colors are for distinguishing between different words.

Furthermore, we conducted a quantitative analysis of word frequency. "Climate" was found in all RHESSys-related literature of 75.6%, "Soil" was 58.9%, "Forest" was 72.2%, "Ecosystem" was 70%, "Carbon" or "Nitrogen" was 38.9%, "Drought" or "Fire" was 24.4%, "Design" or "Management" was 26.7%, "Urban" or "Road" or "Impervious" was 14.4%, while "Sustainability" or "Ecosystem Service" was 5.6%. For detecting the trend of RHESSys research topics, we analyzed the occurrence frequencies of the above words in the published literature every 5 years (Figure 5). Among them, "Climate" and

"Soil" showed the highest frequency, indicating that climate change and soil are common topics in RHESSys studies. Interestingly, "Ecosystem" "Forest" "Carbon" and "Nitrogen" showed a higher frequency of occurrence during the earlier periods while showing a downward trend over the recent periods. This result indicates that topics solely focused on the biogeochemical cycle of forest ecosystems were less popular in recent years. Notably, the studies related to urbanization and land management, represented by the keywords of "Urban" "Design" "Management" and "Impervious", showed an upward trend over time. In particular, the keywords of "Sustainability" and "Ecosystem Service" appeared between 2014 to 2019, and their frequency during this period was 13.2%. It indicates that RHESSys has been applied to integrate ecohydrology into sustainability science in recent years.

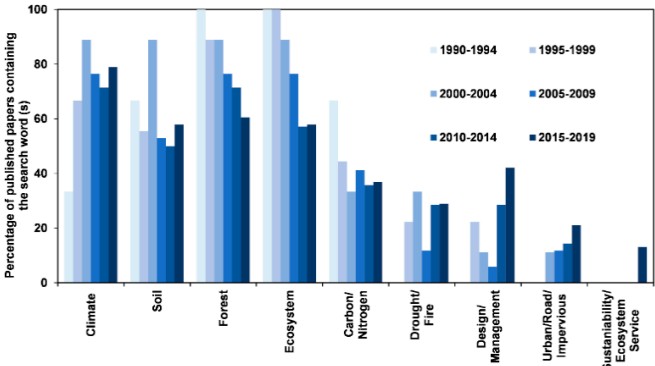

**Figure 5.** The changing trends of selected words and phrases in RHESSys-related papers.

RHESSys has been applied in seven countries on three continents (Figure 6). RHESSys mostly applied in North America, accounting for 85.6% (77 papers); followed by Europe (8 papers, 8.9%), and Asia (5 papers, 5.6%). At a country scale, RHESSys has been mostly used in the United States (71 papers), accounting for 78.9%.

RHESSys also has been applied to various biomes (Figure 6). Mainly: (1) Temperate & Subtropical coniferous forests, (2) Mediterranean forests & Woodlands & Scrub, (3) Tropical & Subtropical broadleaf forest, (4) Temperate grasslands & Savanna & Shrublands, (5) Temperate broadleaf & mixed forests, (6) Deserts & xerophytic shrublands, and (7) Boreal forests. Among them, RHESSys has been mostly applied to the Temperate & Subtropical coniferous forests (28.3%).

Based on the quantitative analysis of the topic words and the classification criteria from the Tague team lab [24], we divided the research topics of RHESSys into six categories: (1) climate change, (2) urbanization, (3) land management, (4) water quality, (5) biogeochemical cycle, and (6) disturbance (Figure 6). Among the six topics, the most popular topic is climate change (30 papers, accounting for 33.3% of the total), followed by biogeochemical cycle (21 papers, 23.3%), disturbance (10 papers, 11.1%), land management (10 papers, 11.1%), water quality (8 papers, 8.9%), and urbanization (8 papers, 8.9%). Climate change has been applied for all the above seven biomes. Ecohydrological responses to urbanization were often studied in coastal areas of the United States and mainly applied in Mediterranean forests & Woodlands & Scrub and Tropical & Subtropical broadleaf forests. Many studies examined water quality for watersheds categorized in Temperate broadleaf & mixed forests in the United States. Studies on land management have been conducted in North America, Europe, and East Asia, in which various biomes exist such as Temperate broadleaf & mixed forests, Mediterranean forests & Woodlands & Scrub, Tropical & Subtropical broadleaf forest, and Temperate grasslands & Savanna & Shrublands. Studies on biogeochemical cycling in various biomes (Temperate & Subtropical coniferous forests, Tropical & Subtropical broadleaf forests, Temperate broadleaf & mixed forests, Deserts & xerophytic shrubland, and Boreal forest) have been implemented in Europe and North America. The topic of disturbance has been commonly studied in drylands, such as temperate grasslands & Savanna & Shrublands, Deserts & xerophytic shrubland, and Mediterranean forests & Woodlands & Scrub.

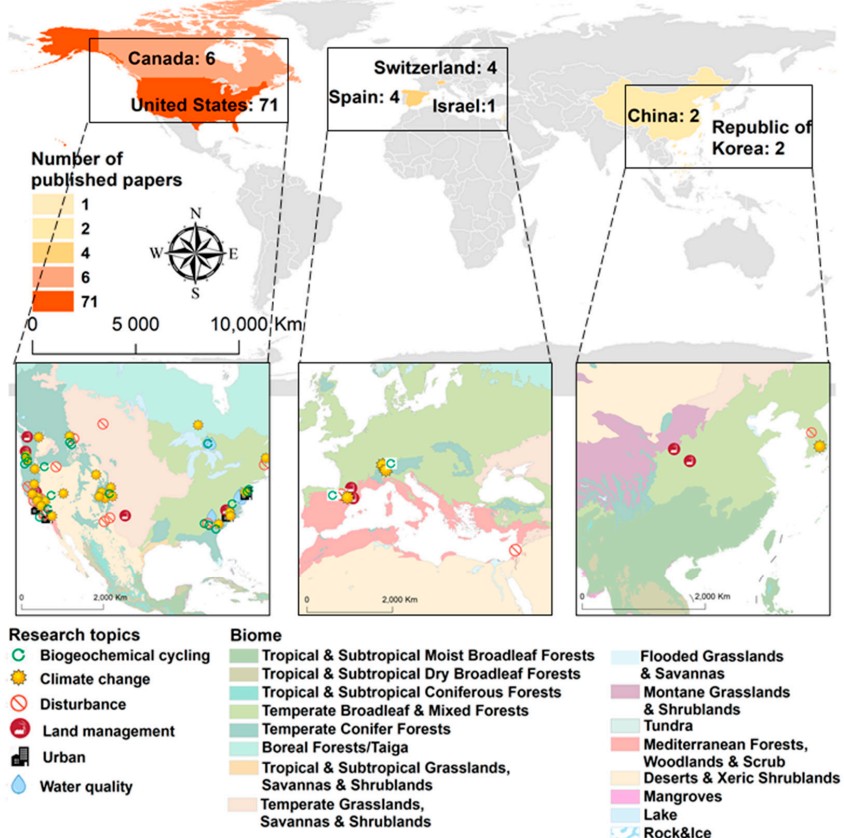

**Figure 6.** The number of published RHESSys-related papers in different countries and the distribution of the research area in global biomes.

RHESSys has been applied to a wide range of spatial and temporal scales as shown in Figure 7. The spatial scale of literature ranged from 0.1 km$^2$ (Yatir Forest, Israel) [25] to 60,000 km$^2$ (South Platte Basin of Colorado, USA) [26], while the temporal scale ranged from 1 to 120 years. However, most studies have focused on relatively small watersheds (less than 100 km$^2$), accounting for 77.3%. In addition, 64.8% of studies have run RHESSys for less than 25 years. Furthermore, 52.3% of the total studies have been applied for study areas less than 100 km$^2$ and time windows less than 25 years account. To date, few studies have applied RHESSys to a large-scale watershed, e.g., there are only two studies for the areas that exceed 10,000 km$^2$ [9,26].

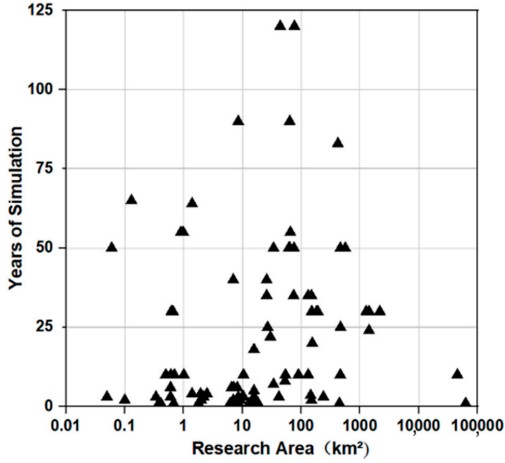

**Figure 7.** Spatio-temporal range of RHESSys-related research.

## 4. Main Application Progress of RHESSys

### 4.1. Climate Change

Climate change directly induces an alteration in the water cycle and threatens the ecosystems' structures and functions, resulting in uncertainties of ecohydrological interactions [27,28]. Understanding the complex ecohydrological response to climate change is crucial for local water resource and ecosystem management. RHESSys is useful on this topic, by combining it with projected climate scenarios, and was performed to evaluate the effects of potential climate change on watersheds and support the regional watershed management [26,29,30].

With meteorological forcings (e.g., rainfall, temperature, and wind speed) from various climate models, RHESSys coupled with other models and comprehensively evaluated the impacts of climate change on the ecosystem and land use (Figure 8). Ecohydrological fluxes incorporated in RHESSys, e.g., streamflow, evapotranspiration, and net primary productivity under climate change, can provide crucial data to support watershed sustainable management under climate change. For example, Son et al. [31] examined the shift in snowmelt, runoff, and evaporation between a snow-dominated basin and rainfall-dominated watershed under projected global warming of 2 or 4 °C, California, USA. Besides, they identified that vegetation structures and soil properties are also crucial factors for ecohydrological responses to climate change. Lopez–Moreno et al. [32] suggested water supply regulation measures for local dams based on streamflow projections for the upper basin of the Spanish Aragon River. Bart et al. [33] evaluated the impact of climate and vegetation changes on the streamflow of Sierra Nevada Basin, California, and projected an increase in the annual streamflow mainly due to climate change and the conversion of primary vegetation type from forest to shrub in the future.

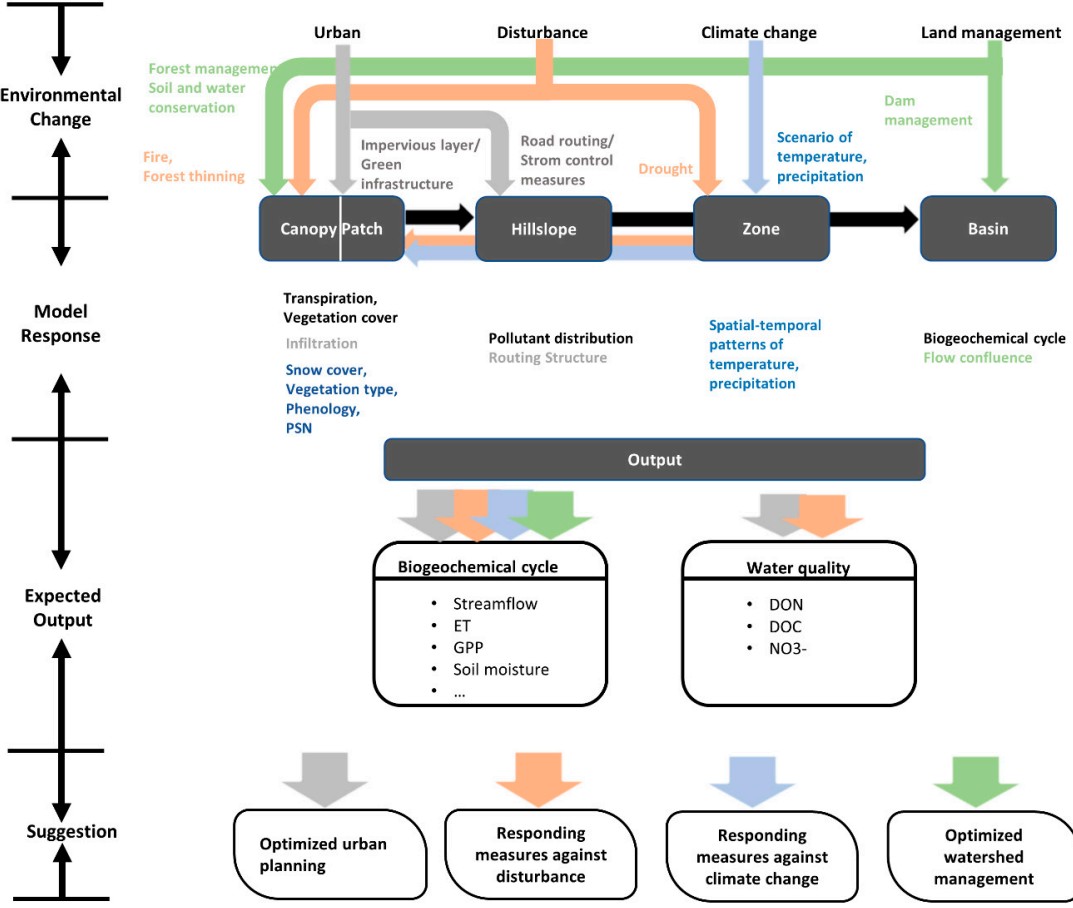

**Figure 8.** Conceptual framework of RHESSys research for different topics.

*4.2. Disturbance*

Disturbances such as wildfires, deforestation, and drought are indispensable factors that regulate the ecohydrological and biogeochemical processes [34]. Fire and forest-thinning effects directly affect the ecological process by reducing the canopy area and the litter layer. Drought induces huge water stress on plant growth, resulting in a considerable influence on ecohydrological processes [35,36].

In many eco-hydrological simulations, disturbance is not included in the model process but is an exogenous force, which rarely depicts the bidirectional feedbacks between watersheds and the disturbance events [4]. However, RHESSys has been used to study the impacts of these disturbances (i.e., fire, drought, and forest thinning) on the ecohydrological process at a watershed scale by setting corresponding disturbance events in the simulation process. A few studies have coupled with a disturbance model, e.g., WMFire (Figure 8). For instance, Kennedy et al. [4] coupled RHESSys with the WMfire model to simulate the impacts of fire on the ecohydrological cycle for two basins located in New Mexico and Oregon. This study successfully linked eco-hydrological outputs with the fire spreading process, which enables further studies on bidirectional interactions between fire and watersheds. Saksa et al. [37] simulated the effects of forest thinning on water resources balance for different watersheds located at the Sierra Nevada Mountains in California. They demonstrated that the forest thinning may cause an increase in annual streamflow, and watersheds that receive a higher rainfall have a more robust response. Hwang et al. [38] proved that RHESSys provides more accurate simulations to evaluate the impacts of drought on the carbon and water cycle in mountain forest areas of East Asia compared with MODIS GPP products that employ the radiation-use efficiency concept.

*4.3. Urbanization*

The imbalance between regional water supply and demands in urbanized watershed urges eco-hydrological simulations to integrate the urban biogeochemical cycle [39]. RHESSys is capable of simulating urbanization and its ecohydrological effects by not only integrating impervious layers into vertical hydrological processes but also considering the effects of road and green infrastructures on convection [40,41] (Figure 8). Compared with other ecohydrological models (e.g., SWAT) that treat the urban area as a single entity, RHESSys especially focuses on the internal urban biogeochemical cycle. For instance, RHESSys has considered the change of water routing by roads and drainage networks, which potentially alters the transpiration, runoff and delivery of water to vegetations [42,43]. RHESSys is capable of identifying a cause–effect relationship between stressors and responses in a watershed, consequently, guiding more targeted managements and monitoring strategies for urbanized basins [44].

Most studies specified urbanization by impervious layers, green infrastructures, and stormwater control facilities, which are modeled by RHESSys to evaluate the interactions with ecohydrological processes. Impervious layer and drainage networks may alter the natural processes in infiltration, transpiration, and lateral flow distribution, resulting in changes in vegetation water usage, streamflow, and other ecohydrological components in a basin. However, a green infrastructure could alleviate these effects. Applying RHESSys for urbanized areas in Charlotte, USA, for example, Bell et al. [44] found that stormwater management measures allowed the basin to hold nitrogen during warm months while causing net nitrogen output during cold months. Shields and Tague [43] found that an increase in impervious layers in Santa Barbara cities may cause a shortage of local vegetation water usage. Rai et al. [45] coupled a green infrastructure design into the RHESSys to assess the best urban green infrastructure design for stormwater control. These studies demonstrated that RHESSys is useful in providing constructive support for urban planning.

*4.4. Water Quality*

The global environment has been threatened by water pollution driven by soil erosion, agricultural fertilizer, municipal wastewater, and atmospheric sedimentation [46]. RHESSys is a physical-based model that simulates the regional distribution of nitrogen and carbon, e.g., the nitrogen

export and dissolved organic carbon [2]. Moreover, RHESSys is a useful tool for the temporally and spatially investigations of the sources of water pollution by quantitatively simulating various processes and elements related to water quality (Figure 8). Especially, RHESSys can incorporate a drainage network, which provides valuable information for urbanized watersheds [44].

Many studies have applied RHESSys to investigate the combined effects of climate change, urbanization, and disturbances on the spatio-temporal variation of water quality. RHESSys mainly simulates water pollutants of dissolved nitrogen, dissolved organic carbon, ammonia nitrogen, and nitrate–nitrogen (Figure 8). RHESSys has been applied to simulate the dissolved organic nitrogen distribution [47] and the soluble organic carbon distribution [48] to support local regulation of water quality. Previous studies also suggested that the model performance can be improved by coupling with groundwater and phenology sub-models.

### 4.5. Land Management

As a watershed usually consists of a number of infrastructures such as water and soil conservations, reservoirs, and irrigation infrastructures, regional land management measures are essential to local water resources management and have a significant impact on regional ecohydrological processes [49,50]. As a comprehensive process, land management is often intertwined with other factors such as climate, vegetation, and land-use change. However, RHESSys has a hierarchical structure that simulates various combinations of scenarios (e.g., climate, land use) at individual scales and thus is enabled to provide land management modeling and its far-reaching impacts on the local basins. Moreover, RHESSys is suitable for coupling with other models, which provides opportunities to study specific land-management measures, such as reservoir construction [6,9,17].

Land management topics usually incorporated climate change and land-use scenarios into the RHESSys model to investigate potential changes in watershed runoff and vegetation productivity and subsequently to evaluate the sustainability of local water resources management (Figure 8). For instance, Martin et al. [6] explored the impacts of future forest management, urbanization, and climate change on hydrologic responses in watersheds in the southeastern United States. The results showed that climate change was the major factor that affects streamflow while the impacts of climate change varied with forest management policies and urbanization levels. Peng et al. [9] investigated the effects of vegetation change with soil and water conservations on runoff in the Jinghe basin of China and concluded that local soil and water conservation measures reduced the average annual flow by 8% and consequently reduced the local soil erosion. Zabalza–Martinez et al. [8] examined the impacts of reservoir management measures on streamflow under future climate and land-use scenarios and suggested adaptations for more efficient watershed water resources management.

### 4.6. Biogeochemical Cycle

The regional biogeochemical cycle closely interacts with the eco-hydrological processes, which are vital processes of matter transportation between soil, water, and atmosphere [51]. As RHESSys has an explicit and dynamic mechanism for the biogeochemical cycle, it has often been used to analyze the impacts of external environmental changes (e.g., climate change and land-use change) on the biogeochemical cycle (Figure 8). Zierl et al. [52] demonstrated that RHESSys can simulate the carbon–water cycles for different forest ecosystems in Europe. Moraels et al. [53] compared the performance between RHESSys and the other three land process models for diverse forest ecosystems in Europe. RHESSys outperformed the others in both the boreal forest and temperate conifer biomes while overestimating the net carbon emission during winter. Zierl et al. [54] explored changes in the carbon cycle for five basins located at different climate zones in the European Alps under multiple climate scenarios. They found that global warming may increase carbon sequestration for all five basins over the first half of the 20th century, while the low-altitude regions may become a carbon source and release carbon continuously over the second half of the 20th century.

## 5. Calibration, Validation, and Uncertainty Analysis of RHESSys

### 5.1. Calibration

Calibration is essential for accurately simulating the regional characteristics of ecohydrological processes [2]. A set of sensitive parameters can be filtered based on their specific physical meanings and contributions to key outputs. Along with empirical tests, four primary calibration parameters have been identified by the model developers: m (Decay of saturated hydraulic conductivity with saturation deficit), Ksat0 (Saturated hydraulic conductivity at the surface), gw1 (Groundwater bypass flow), and gw2 (Groundwater drainage rate) [2]. Other parameters can be considered corresponding to research objectives and key processes. This study further divided those parameters into four groups corresponding to the processes each parameter contributes: (1) soil, (2) snow, (3) vegetation, and (4) water quality.

Table 2 illustrates the primary parameters, target variables, criteria, calibration methods, and representative previous studies for the four groups categorized in this study. Various types of calibration methods, data, and criteria have been applied for the four groups. Soil-related parameters closely related to the hydrological process are often calibrated to be adapted to observed runoff data based on various performance criteria such as NSe (Nash-Sutcliffe efficiency); LogNSe (Nash–Sutcliffe efficiency with logarithmic values); RMSE (Root mean square error); PBIAS (Percent Error, percent volume error and focuses on flow bias); and RSR (Ratio of the root mean square error to the standard deviation of measured data). Many studies have employed a multi-objective function that combines multiple criteria in the calibration process [33,55]. Snow-related parameters are often necessary to calibrate for snow-dominated watersheds. Previous studies have often estimated and calibrated the snow-related parameters based on $R^2$ (determination coefficient) for snow depth, a snow-water equivalent. For example, Son et al. conducted the calibration of snow-related parameters by comparing measured snow depth data and modeled snow-water equivalent, which results in an $R^2$ of 0.87 and 0.70, implying that the model reproduced the real snowmelt process [31,37]. For studies that require detailed ecological outputs, vegetation-related parameters are manually calibrated with remote sensing or field-observed Leaf Area Index (LAI) data [18,38]. For the SCM (Stormwater Control Measures) sub-model developed by Bell et al. [56], 12 water-quality-related parameters contribute to simulating the carbon and nitrogen process in river channels. The calibration procedure starts with generating a certain number of parameter sets and runs the simulation with each set; only the parameter sets that produce a Kolmogorov–Smirnov D < 0.2 for both $NO_3$ and $NH_4$ will be accepted and used for further simulations.

RHESSys employs a Monte Carlo method as a calibration method, referring to GLUE (Generalized Likelihood Uncertainty Estimation) [57,58], not only to analyze the uncertainties of parameters but also to produce optimized parameter sets [9,33,55]. The SCM sub-model also provides a Latin hypercube sampling approach as an alternative. However, the automated calibration method is only available for soil-related parameters and water-quality-related parameters while vegetation-related or snow-related parameters need to be calibrated manually [18,55].

**Table 2.** The details for calibrating RHESSys in existing studies.

| Category | Parameters | Description | Observed Data | Criteria * | Methods | References |
|---|---|---|---|---|---|---|
| Soil | m | The decay rate of saturated hydraulic conductivity with soil depth | Streamflow | NSe, LogNSe, RMSE, PBIAS, RSR | Monte Carlo, GLUE | [9,33,37,55,59] |
| | Ksat0 | The saturated hydraulic conductivity at the soil surface (both dimension) | | | | |
| | gw1 | Groundwater bypass flow, dimensionless | | | | |
| | gw2 | Groundwater drainage rate, dimensionless | | | | |
| | psi | Soil pore-size index, dimensionless | | | | |
| | psi_air_ entry | Soil air-entry pressure, dimensionless | | | | |
| | soil depth | Maximum soil dept, dimensionless | | | | |
| Snow | Lapse_rate | The lapse rates for the daily maximum and minimum air temperature | Snow depth, SWE, Streamflow | $R^2$ | Manually adjusted | [31,37,55] |
| | max_snow_tem | Temperature threshold values for the partition of snow and rain in the total precipitation | | | | |
| | temcf | An empirical temperature melt coefficient (accounting for snowmelt due to latent and sensible heat) | | | | |
| Vegetation | phenology date | Number of days for leaf out period and number of days for litterfall period | LAI | RMSE, Literature-based value | Manually adjusted | [18,38,54] |
| | Q10 | Maintenance respiration (The proportional change in respiration per 10C rise in temperature) | | | | |
| | epc.flnr | Ration Leaf nitrogen in Rubisco to leaf nitrogen | | | | |
| | epc.proj_sla | Specific leaf area | | | | |
| Water quality | kg | Base growth rate of chl-a in algae | DON, DOC fluxes | Kolmogorov-Smirnov D test | Monte Carlo, GLUE, Latin hypercube sampling (LHS) | [44,56] |
| | kd | Base death rate of chl-a in algae | | | | |
| | kr | Base respiration rate of chl-a in algae | | | | |
| | vs | Settling rate of algae as chl-a | | | | |
| | ksn | Half saturation concentration of nitrogen | | | | |
| | ksp | Half saturation concentration of Phosphorous | | | | |
| | P | Phosphorous concentration in the SCM | | | | |
| | kpn | Constant of preferential $NH_4$ uptake, over $NO_3$ | | | | |
| | qg | Constant for kg dependency on temperature | | | | |
| | qd | Constant for kd dependency on temperature | | | | |
| | qr | Constant for kr dependency on temperature | | | | |
| | Is | Optimum radiation level for algae growth | | | | |

* NSe: Nash–Sutcliffe efficiency; LogNSe: Log Nash–Sutcliffe efficiency; RMSE: Root mean square error; PBIAS: Percent Error, percent volume error and focuses on flow bias; RSR: Ratio of the root mean square error to the standard deviation of measured data.

### 5.2. Verification and Uncertainty Analysis

As shown in Table 3, the calibrated parameters that have been validated with the observed data are divided into five groups: runoff, snowmelt, soil, vegetation, and water quality. Among them, the observed runoff data have been used the most in literature for validation based on mainly NSe, LogNSe, PBIAS, or $R^2$ between simulated and observed streamflow (yearly, monthly, or daily) [60–62]. Other performance measures may be necessary for specific study objectives and interests. For instance, Sanford et al. [63] focused on Peak flows and annual fluctuations of streamflow for the Batchawana basin, Canada. This is besides using MLE (Maximum likelihood Estimate) for a comprehensive evaluation of simulation results with multi-parameters combination sets [64].

In addition to the observed streamflow, other observed datasets available in a watershed provide the informative characteristics of hydrologic and ecological processes. For example, applications for small basins often need to validate the simulated soil water distribution. Boisrame et al. [65] evaluated the correlation between observed and simulated soil moisture contents. Measured snowmelt and snow depth data have been used to verify the simulation results for snow-dominated basins [37,66]. Furthermore, plant ecological processes have been validated with observed transpiration, net primary productivity, gross primary productivity, photosynthesis rate, or leaf area index, which are driven by remote-sensing products, site data, and field measurement. Quantitative analysis has been conducted with correlation analysis and other statistical measures. Kim et al. [5] conducted a correlation analysis between site-monitored and simulated transpiration and found that the coefficients ranged from 0.69 to 0.91 and other performance measures were above an acceptable level (the total error percentage of 14.6% and the mean square root error ranging from 0.35 to 1.15). Given that the observed data are insufficient for quantitative analysis, qualitative comparisons can be conducted. For example, Peng et al. [64] evaluated the accuracy of simulated weekly net primary productivity (NPP) (2.85 gC m$^{-2}$ day$^{-1}$) by comparing it with field-measured NPP (2.81 gC m$^{-2}$ day$^{-1}$). For water quality studies, DOC and $NO_3$ have been used to validate simulations. Son et al. [31] performed WRTD (Weighted Regression Model) statistical analysis between site-monitored and simulated dissolved organic carbon to calculate the NSe and LogNSe. Bell et al. [44] evaluated the performance in the monthly $NO_3$ concentration and showed that the correlation coefficient and NSe were 0.82 and 0.64, respectively.

**Table 3.** The details for validating RHESSys in existing studies.

| Category | Observed Data | Source | Time Resolution | Criteria * | Methods ** | References |
|---|---|---|---|---|---|---|
| Streamflow | Streamflow | Gauge measurement | Daily, Monthly, Annual | NSe, LogNSe, PBIAS, $R^2$ | Statistical analysis | [61] |
| | | Gauge measurement | Daily | Peak flow error, NSe, Flow variability, $R^2$ | Statistical analysis | [63] |
| | | Gauge measurement | Daily, Monthly, Annual | NSe, PBIAS | MLE | [64] |
| Soil | Soil moisture | Field measurement | Daily | R | Correlation analysis | [65] |
| Snow | Snowmelt | Field measurement | Daily | $R^2$ | Correlation analysis | [37] |
| | Snow depth | Field measurement | Daily | $R^2$ | Correlation analysis | [54] |
| Vegetation | ET | Flux tower measurement | Daily | $R^2$ | Correlation analysis | [59] |
| | | Flux tower measurement | Daily | RMSE, PBIAS, $R^2$ | Correlation analysis | [5] |
| | GPP | Remote sensing | Monthly | $R^2$ | Correlation analysis | [59] |
| | | Flux tower measurement | Daily | RMSE, PBIAS, $R^2$ | Correlation analysis | [5] |
| | NPP | Field measurement | Daily | N/A | Qualitative | [64] |
| | | Field measurement | Annual | N/A | Qualitative | [67] |
| | | Remote sensing | Annual | Mean error | Statistical analysis | [9] |
| | PSNet | Remote sensing | Daily | $R^2$ | Correlation analysis | [59] |

**Table 3.** *Cont.*

| Category | Observed Data | Source | Time Resolution | Criteria * | Methods ** | References |
|---|---|---|---|---|---|---|
| Vegetation | Transpiration | Field measurement | Daily | R | Correlation analysis | [64] |
| | LAI | Field measurement | Daily | N/A | Qualitative | [68] |
| | | Remote sensing | Daily | N/A | Qualitative | [9] |
| Water quality | DOC | Gauge measurement | Daily, Monthly, Annual | NSe, LogNSe | WRTD | [48] |
| | NO$_3$ | | Monthly | NSe, R | Statistical analysis | [44] |

* Flow variability: S80 [(90th percentile—10th percentile)/50th percentile] were compared with the S80 of observed flow characteristics using linear regressions; Peak flow error: Term to establish the mean square error around peak flows where 0 represents no error and 1 represents the complete error; ** MLE: Maximum likelihood estimate by combining model estimates from each parameter set, weighted by their performances; WRTD: A weighted regression model and uses time, discharge, and season to predict the concentration and fluxes for stream water quality analysis.

The uncertainty of RHESSys is mainly sourced from input data, the model structure and algorithms, and the parameters (Table 4). First, RHESSys requires various observed data to be forced for simulating natural processes. However, the observed data are often insufficient to be used for modeling with regard to data length and quality. Therefore, multi-source data have been incorporated to compensate for the lack of monitored data such as rainfall, runoff, and soil data in watersheds [6,34,37]. Besides, information on the carbon and nitrogen pools is limited. To address this, RHESSys assumes that the vegetation, carbon, and nitrogen pools within the study area reach the equilibrium to continuously run the model as a dynamic system. Then, RHESSys simulates vegetation change, water redistribution, and soil biogeochemical cycles until the equilibrium or the goal state set is reached (so-called 'spin-up'). Given the state produced by the spin-up process as an initial condition, RHESSys simulates system responses during a subsequent period [2]. Nevertheless, the spin-up process still has uncertainties and needs to be improved by incorporating remote-sensing data to accurately set the goal initial state. The resolution of DEM also affects the confluence in model simulations. In other words, the coarser the resolution of DEM data, the greater the deviation from observed data may be induced, and vice versa [55].

The uncertainty of RHESSys also is sourced from its structure and algorithms. RHESSys interpolates climate station data to target points within a watershed, which provides the regional rainfall and temperature patterns. However, the distribution of regional temperature and rainfall may be inaccurate for data-sparse or complex topography areas. Although RHESSys does not model the dynamic process of vegetation succession and structures, phenology models have been coupled to rectify the flaws [66]. Moreover, RHESSys does not consider the in-stream processes of carbon and nitrogen, leading to overpredicting the water pollution outputs. Bell et al. [56] developed a SCM submodule to simulate the chemical processes of carbon and nitrogen in river channels. However, the SCM submodule requires more detailed input data such as the distribution of urban drainage networks. In addition, RHESSys assumes all channelized flows are discharged from a basin within a day. However, the assumption cannot reflect the real confluence process in a large basin. Peng et al. [9] developed an in-stream routing submodule for stream networks within basins to expand the applicability of RHESSys to large basins.

The parameters of the model also bring uncertainties to the simulation results. Most studies generalized soil and vegetation parameters or employed empirical values suggested by previous studies [69] due to limited data regardless of heterogeneity in spatial soil and vegetation types. Detailed field observation or multi-source data are critical for reducing uncertainty [66,70]. Besides, detailed fieldwork is necessary to determine parameter values to produce high accuracy outputs by more efficient and accurate calibrations and consequently reducing uncertainties.

**Table 4.** Sources and corresponding solutions of uncertainties in RHESSys.

| Source | Specific Sources | Solutions | References |
|---|---|---|---|
| Input Data | Coarse-resolution of DEM | Fine-resolution DEM data | [55] |
| | Lack of detailed precipitation, gauge, soil, and other basic data | Multi-source data acquisition | [6,34,37,71] |
| | Lack of carbon fluxes and pool data | Spin-up strategy, and integrate remote sensing data | [34] |
| Model structure and algorithms | Interpolation strategy of air temperature and precipitation | Explicitly incorporate spatial characteristics of surface metrological variables | [7,37,54,71–73] |
| | Without the plant migration and structure change | Couple vegetation dynamic models | [66] |
| | Without the in-stream process | SCM sub-model | [2,56] |
| | Scale problem | In-stream routing, and adapted landscape partitioning strategy | [9] |
| Parameters | Simplification of vegetation variability | Multi-source data assimilation | [66,74] |
| | Empirical parameters | Detailed filed studies, Calibration | [54,66,71,75] |

## 6. Future Perspectives of RHESSys

### 6.1. Key Challenges

The application of RHESSys has been hindered by the complexity and availability of parameters, and the requirement of detailed data. As a physical process-based model, RHESSys simulates the ecohydrological processes at the expense of involving substantial parameters and data support, which impedes the application of RHESSys and induces more uncertainties [43,71]. Son et al. [31] found difficulties in determining snow and soil parameters for dissolved carbon simulations, which may cause a bias in model outputs. Martin et al. [6] pointed out the lack of detailed rainfall intensity and urban drainage data, which may induce underestimating the peak streamflow. Although empirical parameters have been provided for some biomes [2,69], the model users often need to modify the ecological parameters to cope with localized vegetations [76].

The calibration approaches incorporated into RHESSys also need to be improved. Currently, RHESSys employs the Monte Carlo method that optimizes the parameters by randomly sampling paired parameter groups and picking out a group with the best performance. This approach usually requires tremendous computational resources. Furthermore, the automatic optimization is available only for soil-related and water quality-related parameters in RHESSys. Although Reyes et al. [77] applied a Latin super-square sampling method to optimize carbon-allocation parameters, most studies have calibrated vegetation-related parameters manually. Thus, the automated and more systematic and efficient calibration methods for vegetation-related parameters are needed [38,66].

As RHESSys has initially been designed for small-scale basins [2,9], application for a large-scale basin is a challenge. Since large watersheds have strong spatial heterogeneity in the ecosystem and usually include data-sparse areas, it is often challenging to obtain sufficient data and parameters to be used for RHESSys, resulting in high uncertainties. Besides, the applications for a large-scale basin require high-performance hardware to perform huge computational tasks. Therefore, the current applications often simplify vegetation types and degrade the spatial resolution to reduce the complexity of modeling for large basins [9,26].

RHESSys has been applied for a variety of fields such as climate change and land management. However, the current studies have mainly focused on natural systems in watersheds while few studies have taken into account socio-economic systems. Few studies have paid attention to linking the outputs of the carbon, nitrogen, and water simulations from RHESSys to the regional water supply–demand balance, ecosystem services, or human well-being. RHESSys has a lot of potentials to quantify ecosystem services more accurately and to be a useful tool for studies on watershed sustainability. However, there are few relevant studies in the literature [25,78].

*6.2. Future Directions*

RHESSys needs to enhance the abilities of data collection for improving the simulation capabilities of regional water resources and land management [79]. A single data source may often result in the overfitting phenomenon of parameters, leading to unreliable simulations and predictions [73]. In this sense, multi-source data composition and data assimilation methods can be an alternative to be employed into RHESSys. For instance, Sakas et al. [37] effectively improved the calibration efficiency by incorporating multi-sources data such as remote-sensing products, ground observations, and field measurements. Hanan et al. [34] also incorporated remote sensing data to set a goal state in the spin-up process, which enhanced the reliability and accuracy of model outputs. Moreover, a more comprehensive parameter library can be built from various applications in parameter localization and advanced remote-sensing technologies.

A number of calibration methods have been developed and applied for hydrologic models such as simulated annealing (SA) [80], genetic algorithm (GA) [81] and shuffled complex evolution method (SCE-UA) [82]. Therefore, RHESSys needs to incorporate the most suitable calibration methods in the future to improve calibration efficiency.

In addition, RHESSys is necessary to be further adapted to large-scale basins. Over the last years, ecohydrological simulations at a large-scale have received attention more and more as climate and land-use change have intertwined with ecohydrology [83]. As the current version of RHESSys may not be suitable for simulating a large-scale basin, the model structure and some mechanisms need to be modified to adapt to a large-scale basin. Moreover, a parallel computation module is also very useful to reduce computational burdens.

As human activities have intensively increased in watersheds, previous RHESSys-related studies have investigated the impacts of human activities on the ecohydrological process, such as urban planning, agricultural irrigation, soil and water conservation, and reservoir construction [8,9,44]. It is necessary to project the impacts of human activities on sustainability in a watershed. Consequently, RHESSys provides a useful tool to quantify watershed ecosystem services and to assess regional sustainability, resulting in promoting sustainable development for watersheds.

## 7. Conclusions

Over the last 30 years, RHESSys has been applied for ecohydrological studies, mainly for seven biomes around the world, such as temperate conifer forests, Mediterranean forests, woodlands & Scrub, and temperate grasslands. This study demonstrated that the number of published papers and citations on RHESSys increased over time, indicating that RHESSys have been paid attention by many researchers. RHESSys has generally been applied for basins smaller than 100 km$^2$ with time windows less than 25 years. To date, RHESSys has been used for mainly six topics, such as climate change, land management, urbanization, disturbance, water quality, and biogeochemical cycle. Most studies have focused on the ecohydrological responses to climate change, while in recent years, there has been a rapid increase in land management and urbanization.

This study also proposed challenges for RHESSys: data acquisition, calibration methods, and applicable basin scales. As RHESSys requires a number of parameters, it is a challenge to acquire all of the data required for modeling. Additionally, the current calibration method is relatively simple but requires a huge computational burden. Lastly, RHESSys is not suitable for simulating at a large spatial scale during a long-time span, as the data and parameter accessibility are limited.

In the future, multi-source data and data assimilation methods are required to improve the ability of data collection and subsequently to enhance the capabilities of RHESSys for water resources and land management. This study also suggested incorporating more advanced calibration methods into RHESSys. Furthermore, the model structure needs to be modified to improve the applicability of RHESSys to large-scale basins and to reflect the human–nature interactions for more sustainable management of watersheds.

**Supplementary Materials:** The following are available online at http://www.mdpi.com/2073-4441/12/10/2878/s1.

**Author Contributions:** B.C. drafted the manuscript. Z.L. conceived and guided this study. C.H. and H.P. gave important advice on methodology and providing suggestions on the revision of the manuscript. P.X. and Y.N. offered advice on the revision of figures and tables. All authors have read and agreed to the published version of the manuscript.

**Funding:** This work was funded by the National Natural Science Foundation of China (Grant No. 41871185) and the Second Tibetan Plateau Scientific Expedition and Research Program (Grant No. 2019QZKK0405). It was also funded by the National Natural Science Foundation of China (Grant No. 41971271&41971270) and a project from the State Key Laboratory of Earth Surface Processes and Resource Ecology, China.

**Acknowledgments:** We want to express our respect and gratitude to Yihang Wang, Yanmin Yang, and Yihua Dai for their help.

**Conflicts of Interest:** The authors declare no conflict of interest.

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
