# Peer review of "The Regional Hydro-Ecological Simulation System for 30 Years: A Systematic Review"

_water, doi:10.3390/w12102878_

Round 1

Reviewer 1 Report

Please see the comments in the pdf file.

Reviewer 2 Report

The authors provide comporehensive review of the essense and applications of the RHESSys model which is widely used for modelling the interactions between ecological and hydrological processes. The authors clearly showed the evolution of the model as well as shortcomings and desirable future development. Various applications are demonstrated. Bibliometric analysis demonstrates the changes in ways of the model application which reflect shifts in scientific priorities during almost 30 years long period. Demonstration of the multi-scale essense of the model looks  fine. The model is compared with the other ones. This is useful for the researchers in order to make their choice based on the relevant objectives and scale. It is very important that the paper provides not the review of applications only but review of calibration, validation, and uncertainty analysis as well. Future perspectives, as they are seen by the authors, are in compliance with the present-day trends of ecohydrology and landscape ecology. The list of references and 90 RHESSys-related papers will be helpful for the readers who already uses or plans to use the model or to compare its effeciency with the other models. 

Reviewer 3 Report

I generally welcome this review of a process-based ecohydrological model. Since I am not an expert of RHESSys, this overview added to my knowledge in many ways. However, I found many issues regarding the style and presentation of the review. Overall, the manuscript is too long for its content and too general for its focus on a highly sophisticated method. Condensing, clarifying and adding detail are necessary. Please see the attached file for details, but my criticism generally falls into one of the following broader problems.

1. Most of the subject treatment remains very general and indicative, instead of clarifying how this specific model has added to knowledge base. Yet, it is not possible to understand almost anywhere whether other modelling approaches would have also been useful. Please add such comparisons!

2. The style suffers from wordiness. I have tried to mark the places or phrases that should be deleted or condensed. I also did not like the superlatives to describe the model that start from the 1st sentence of Abstract; such usage could only be justified it such superior performance of RHESSys had been actually documented (followed by refs). 

3. The analyses are very much based on just counting or listing papers. Of these, the listing of research foci (pp. 3-10) is based on word counting, which is  rather a visual tool than a research method (Fig. 4). Tables 2-3 are probably not needed; the former lists papers based on citations that accumulate over time and may reflect usage but not necessarily the quality. Please note also that Fig. 2 requires formal copyright permissions, since the subgraphs are almost copy-paste from the original sources. Etc. I think the list of 90 papers is really needed as an Appendix, but I would condense the paper more toward actual analysis and synthesis.

Round 2

Reviewer 3 Report

I have not re-evaluated this manuscript and was happy to see that the authors carefully considered my comments and made efforts to improve. I only found that you have not deleted Fig. 5a as you claimed in your point 7: it still duplicates the text (currently at lines 179-183). For conciseness I thus repeat the suggestion to retain the text percentages and to delete part (a) of Fig. 5.

Apart from that, I only have a comment that please make a careful spelling check - there are many small errors! Below just examples that I found from a short section of the paper (did not make it for the whole text):

line 104-105 consider rephrasing "in which each level represents distinct hydrological and ecological processes"

line 125 delete "when"

line 130-135 very long sentence, consider splitting

line 135, 148 et al. (full stop missing)

line 138 relies -> uses

line 141 simulations

line 153 was -> were

line 154-155 consider rephrasing "After reading the full texts, a total of 90 papers related..."

line 157 delete "that"

line 163 has been continuously raised -> has steadily grown

Fig. 6 misspelling Switherland -> Switzerland

Line 245 has proved -> is
